# Sex-Specific Effects of Plastic Caging in Murine Viral Myocarditis

**DOI:** 10.3390/ijms22168834

**Published:** 2021-08-17

**Authors:** Katelyn A. Bruno, Logan P. Macomb, A. Carolina Morales-Lara, Jessica E. Mathews, J. Augusto Frisancho, Alex L. Yang, Damian N. Di Florio, Brandy H. Edenfield, Emily R. Whelan, Gary R. Salomon, Anneliese R. Hill, Chathuranga C. Hewa-Rahinduwage, Ashley J. Scott, Henry D. Greyner, Frank A. Molina, Merci S. Greenaway, George M. Cooper, DeLisa Fairweather

**Affiliations:** 1Department of Cardiovascular Medicine, Mayo Clinic, Jacksonville, FL 32224, USA; Bruno.Katelyn@mayo.edu (K.A.B.); macomb.logan@mayo.edu (L.P.M.); carolina.moraleslara93@gmail.com (A.C.M.-L.); bkjessica12@yahoo.com (J.E.M.); alex.lingyun.yang@gmail.com (A.L.Y.); diflorio.damian@mayo.edu (D.N.D.F.); whelan.emily@mayo.edu (E.R.W.); salomon.gary@mayo.edu (G.R.S.); hill.anneliese@mayo.edu (A.R.H.); 2Department of Clinical and Translational Science, Mayo Clinic, Jacksonville, FL 32224, USA; 3Department of Immunology, Mayo Clinic, Jacksonville, FL 32224, USA; 4Department of Environmental Health Sciences, Johns Hopkins Bloomberg School of Public Health, Baltimore, MD 21215, USA; afrisper@gmail.com (J.A.F.); ajscott6@wisc.edu (A.J.S.); hdga92@gmail.com (H.D.G.); frankmolina92@gmail.com (F.A.M.); merci.clifford@gmail.com (M.S.G.); george-cooper@uiowa.edu (G.M.C.); 5Department of Cancer Biology, Mayo Clinic, Jacksonville, FL 32224, USA; edenfield.brandy@mayo.edu; 6Department of Chemistry, Sam Houston State University, Huntsville, TX 77340, USA; hrcchinthana2@gmail.com

**Keywords:** bisphenol A, myocarditis, sex differences, endocrine disruptors, coxsackievirus B3

## Abstract

Background: Myocarditis is an inflammatory heart disease caused by viral infections that can lead to heart failure, and occurs more often in men than women. Since animal studies have shown that myocarditis is influenced by sex hormones, we hypothesized that endocrine disruptors, which interfere with natural hormones, may play a role in the progression of the disease. The human population is exposed to the endocrine disruptor bisphenol A (BPA) from plastics, such as water bottles and plastic food containers. Methods: Male and female adult BALB/c mice were housed in plastic versus glass caging, or exposed to BPA in drinking water versus control water. Myocarditis was induced with coxsackievirus B3 on day 0, and the endpoints were assessed on day 10 post infection. Results: We found that male BALB/c mice that were exposed to plastic caging had increased myocarditis due to complement activation and elevated numbers of macrophages and neutrophils, whereas females had elevated mast cell activation and fibrosis. Conclusions: These findings show that housing mice in traditional plastic caging increases viral myocarditis in males and females, but using sex-specific immune mechanisms.

## 1. Introduction

Myocarditis is an inflammatory heart disease that is caused by viral infections including coxsackieviruses (CVB) and SARS-CoV-2, which can lead to acute heart failure, or progress to dilated cardiomyopathy (DCM) and chronic heart failure [1,2,3,4,5,6,7,8]. Of the total reported cases for DCM, up to one-third are induced by myocarditis [9,10]. More men than women develop myocarditis and DCM [11,12]. Additionally, men with myocarditis are more likely to develop cardiac fibrosis than women, and progress to DCM and heart failure [13,14]. Inflammation and fibrosis play significant roles in the cardiac remodelling process. Evidence supports the idea that there is a link between cardiac inflammation and the development of cardiac fibrosis within the perivascular and interstitial spaces of the heart [15,16]. T helper (Th)2 cells, which are associated with the activation and mediation of allergic inflammatory responses, often involving mast cells, were found to present with profibrotic characteristics. The development of fibrosis has been associated with the release of Th2 cytokines that include interleukin (IL)-4, IL-5, and IL-13 [17]. Other proinflammatory cells, such as Th17 cells, have been demonstrated to link inflammation and fibrosis, through the secretion of IL-17A, which was observed to promote hypertension-induced fibrosis, and to promote remodeling and fibrosis that lead to DCM in animal models and patients with myocarditis [18,19,20]. Monocytes also play a large role in the connection between inflammation and fibrosis. Similar to the actions of Th2 and Th17A cells, cytokines that are associated with inflammation and produced by monocytes (TNF-α, IL-6, and IL-1β), have also been observed to be profibrotic [17]. Certain types of cells, such as mast cells and eosinophils that are associated with Th2-type immune responses, are known to promote the development of fibrosis following tissue injury or damage [21]. Previously, we showed that α1-antichymotrypsin, which is released exclusively from mast cells, was upregulated during CVB3 myocarditis and promotes cardiac remodeling and fibrosis [22]. We recently reported a sex ratio of 3.5:1 male-to-female for patients with myocarditis [23]. Additionally, testosterone has been found to increase viral myocarditis in male mice, while 17β-estradiol decreases disease in females [24]. We previously reported that male mice with myocarditis have elevated levels of CD11b, also known as complement receptor (CR)3 that is increased on macrophages, neutrophils, and mast cells, and elevated by testosterone [25,26]. Additionally, we showed that Toll-like receptor (TLR)-4 expression is increased on splenic and heart infiltrating CD11b+ immune cells, including mast cells, during the innate and adaptive immune response during myocarditis [22,25,27]. CR1, which binds complement C3, inhibits its binding to CR3, thereby reducing viral myocarditis, remodeling, fibrosis, and DCM [28]. Mast cells are key cells that are activated by complement and play a critical role in promoting myocarditis, remodeling, and fibrosis during myocarditis, by releasing enzymes such as α1-antichymotrypsin (serpin A3n) that activate cytokines such as IL-1β and matrix metalloproteinases (MMPs), which drive the remodeling that leads to fibrosis and DCM [22]. IL-1β, IL-6, and IL-17A responses are elevated in men with myocarditis/cardiomyopathy, and associated with poor recovery from heart failure [19].

Bisphenol A (BPA) is an endocrine-disrupting chemical that is used in the production of polycarbonate plastics and epoxy resins, and is found in items such as plastic water bottles, plastic food containers, the lining of cans, on thermal receipts, and photocopy paper [29,30,31,32]. Studies have found that people of all ages have detectable levels of BPA, or its metabolized products, in their body fluids [33,34]. BPA has been detected in nearly all patients when assessed in the urine or blood [34,35,36,37,38]. Epidemiological and animal data indicate that increased exposure to BPA worsens cardiovascular diseases, including hypertension [39,40,41], atherosclerosis [42,43,44], myocardial infarct [45], arrhythmias [46,47], and DCM [48], as well as autoimmune and inflammatory diseases [40,41,44,49,50,51,52,53,54,55,56]. A positive correlation has also been made between BPA exposure and the subsequent rise in cardiovascular diseases, such as cardiomyopathy, myocardial infarcts, and congestive heart failure in the US population [48,57,58]. Myocarditis leads to 1/3 of all DCM cases and, therefore, BPA could increase myocarditis which then progresses to DCM. Importantly, an endocrine disruptor, such as BPA, could alter the effect of sex hormones on cardiac inflammation following viral infection in a sex-specific manner. We previously showed that a human-relevant exposure of BPA, administered in drinking water to female mice, increased viral myocarditis to levels similar to males [59]. Few studies have examined the effect of BPA exposure on males [50,60,61]. To our knowledge, no one has examined the effect of BPA on myocarditis in males.

Because viral myocarditis is influenced by sex hormones, we hypothesized that the endocrine disruptor BPA, which we found altered the immune response to myocarditis in females, may play a role in the progression of disease in males. In studies that assessed the pharmacokinetics of BPA, its effect was primarily mediated through the estrogen receptor (ER), but was also found to bind the androgen receptor (AR) as an agonist at a low level (at about 20% of the binding efficiency of testosterone) [62]. BPA has also been found to decrease ERα expression in the spleen of male rats [63]. Similarly, BPA treatment of male human T-cell lymphoblast lines in culture was found to increase ERβ expression using qRT-PCR [64]. In this study, we examined whether BPA exposure from plastic caging could alter CVB3 myocarditis in adult male and female BALB/c mice. We compared the effect of this environmental exposure on viral myocarditis in mice that were given the EPA reference dose of BPA as a positive control.

## 2. Results

### 2.1. Plastic Caging Increases Myocarditis in Male BALB/c Mice during Viral Myocarditis

In order to assess the effect of plastic caging on myocarditis, in this study we examined the effect of exposure to plastics from traditional plastic cages and plastic water bottles (plastic caging), and control glass cages and glass water bottles (glass caging), in 6–8-week-old male and female BALB/c mice with CVB3 myocarditis, using control water (containing no added BPA). The mice received soy-free food and bedding for all the experiments. As a positive control, the male and female BALB/c mice were given 250 μg BPA/L in their drinking water, which is equivalent to an estimated intake of 50 μg BPA/kg body weight (BW), which is equivalent to the EPA reference dose or the maximum daily oral exposure dose that is likely to occur over a lifetime without deleterious effects [65]. The mice who received BPA in their drinking water were housed in glass caging with soy-free food and bedding.

We found that exposure to plastic caging in BALB/c female mice (with no BPA added to the drinking water) did not significantly alter myocardial inflammation histologically, at day 10 post infection (pi), during peak myocarditis (*p* = 0.34) (Figure 1a). However, a BPA exposure of 250 μg BPA/L in the drinking water, which is equivalent to an estimated intake of 50 μg BPA/kg BW [65], significantly increased myocarditis histologically in female mice (*p* = 0.02) (Figure 1a). One-way ANOVA in female mice was significantly different between the groups (*p* = 0.002) (Figure 1a). In contrast, plastic caging significantly increased myocarditis in males compared to glass caging that was assessed histologically (*p* = 0.02), similarly to high-dose BPA (*p* = 0.02) (Figure 1b). The one-way ANOVA in male mice was significantly different between the groups (*p* = 0.02) (Figure 1b). Representative images of the H&E stains of the hearts from the glass, plastic, and 50 BPA (Figure 1c) groups show increased myocarditis (% inflammation) in the plastic and 50 BPA groups compared to the glass group in male mice.

### 2.2. Plastic Caging and BPA Exposure Has No Significant Affect on VP1 Viral Gene Expression in Males or Females during Myocarditis

We found that the early viral gene VP1 of CVB3, indicating viral replication [66], was not upregulated in the heart of female (Figure 2a) or male mice (Figure 2b) during myocarditis at day 10 pi, due to plastic caging or BPA exposure. These data suggest that the increase in myocarditis that was found in male BALB/c mice who were exposed to plastic caging and BPA in their drinking water (Figure 1) was not due to increased virus levels in the heart. The one-way ANOVA was not significantly different between the groups in females (*p* = 0.29) (Figure 2a) or males (*p* = 0.57) (Figure 2b).

### 2.3. Plastic Caging Increases Macrophage and Neutrophil Markers in Males during Myocarditis by qRT-PCR

Exposure to plastic caging in BALB/c females did not significantly increase myocarditis histologically (Figure 1a), or the gene expression of major immune cell markers (CD45, CD11b, F4/80, GR1, CD14, CD3, CD4, and CD8), compared to glass caging, as previously reported in [59]. The only marker that was increased in the females was cKit, which is a marker of mast cells (*p* = 0.04). In contrast, males that had been exposed to plastic caging had significantly increased gene expression of CD11b (*p* = 0.001), GR1 (*p* = 0.005), and CD14 (*p* = 0.006) in the heart during myocarditis using qRT-PCR (Figure 3b,d,e). Plastic caging did not significantly alter CD45 (*p* = 0.49) or F4/80 (*p* = 0.17) gene expression in the heart of the males with myocarditis compared to glass caging (Figure 3a,c). However, all the inflammatory cell markers, except GR1, that were analyzed were significantly increased by exposure to 250 μg BPA/L in the drinking water including CD45 (*p* = 0.009), CD11b (*p* = 0.02), F4/80 (*p* = 0.001), and CD14 (*p* = 0.02) (Figure 3a–c,e). The one-way ANOVA was significantly different between the groups for CD45 (*p* = 0.01) (Figure 3a), CD11b (*p* = 0.002) (Figure 3b), F4/80 (*p* = 0.002) (Figure 3c), GR1 (*p* = 0.008) (Figure 3d), and CD14 (*p* = 0.008) (Figure 3e). Thus, plastic caging increases viral myocarditis in male and female mice, by activating different immune cell populations according to sex.

### 2.4. Plastic Caging Has No Significant Effect on T-Cell Markers in Males and Females during Myocarditis

Exposure to plastic caging in BALB/c mice did not significantly alter the gene expression of markers that were associated with T cells including CD3, CD4, or CD8, compared to glass caging in males (Figure 4) or as previously reported for females in [59]. However, exposure for 2 weeks to 250 μg BPA/L (50 BPA) in drinking water significantly increased the expression of CD3 (*p* = 0.007) and CD4 (*p* = 0.02) in the heart compared to the 0 BPA control water (glass) during CVB3 myocarditis in the males (Figure 4a,b). The one-way ANOVA was significantly different between the groups for CD3 (*p* = 0.01) (Figure 4a) and CD4 (*p* = 0.01) (Figure 4b), but not for CD8 (*p* = 0.28) (Figure 4c).

### 2.5. Plastic Caging Increases Macrophages and Neutrophils in Males during Myocarditis by IHC

Plastic caging had significantly increased the gene expression of CD11b and GR1 in the heart during myocarditis using qRT-PCR (Figure 3b,d). The exposure to 250 μg BPA/L in the drinking water increased the gene expression of CD45, CD11b, F4/80, and CD3 compared to 0 BPA/glass (Figure 3a–c and Figure 4a). To confirm the findings from the RT-PCR gene expression data, immunohistochemistry (IHC) was performed for CD45, CD11b, F4/80, GR1, and CD3. The males that had been exposed to plastic caging had significantly increased positively stained cells for CD11b (*p* = 0.008) and GR1 (*p* = 0.01) in the heart, during myocarditis, using IHC (Figure 5b,d). The exposure to 250 μg BPA/L in the drinking water also increased the positive-stained cells for CD45 (*p* = 0.005), CD11b (*p* = 0.04), F4/80 (*p* = 0.007), GR1 (*p* = 0.001), and CD3 (*p* < 0.0001) compared to 0 BPA/glass (Figure 5a–e). The one-way ANOVA was significantly different between the groups for CD45 (*p* = 0.009) (Figure 5a), CD11b (*p* = 0.01) (Figure 5b), F4/80 (*p* = 0.01) (Figure 5c), GR1 (*p* = 0.01) (Figure 5d), and CD3 (*p* < 0.0001) (Figure 5e). Thus, plastic caging increases viral myocarditis in male BALB/c mice by increasing myocardial macrophages and neutrophils.

### 2.6. Plastic Caging Increases the Mast Cell Marker cKit in the Heart of Females with Myocarditis, but Decreases it in Males

We found that the only immune cell marker in the heart that was significantly increased during myocarditis at day 10 pi by qRT-PCR after exposure to plastic caging in female BALB/c mice was the mast cell marker cKit (*p* = 0.02) (Figure 6a). However, we did not see an increase in cKit expression in the heart after exposure to 250 μg BPA/L (50 BPA) in the drinking water in females compared to 0 BPA/glass (Figure 6a). In contrast, housing male BALB/c mice in plastic caging significantly decreased the expression of cKit (*p* < 0.0001) (Figure 6b), similarly to exposure to 250 μg BPA/L (50 BPA) in the drinking water (*p* = 0.04) (Figure 6b). The one-way ANOVA was significantly different between the groups in females for cKit (*p* = 0.005) (Figure 6a), and for males (*p* < 0.0001) (Figure 6b).

### 2.7. Plastic Caging Increases Pericardial Mast Cell Numbers and Degranulation in Females during Myocarditis

Next, we examined the mast cell numbers and degranulation histologically, and found that plastic caging significantly increased the total number of cardiac degranulating mast cells (*p* = 0.006) (Figure 7b) and the number of pericardial mast cells that were degranulating (*p* = 0.0002) (Figure 7c), compared to the mice who were housed with glass caging. The same result was observed when the mice were exposed to 250 μg BPA/L (50 BPA) in the drinking water compared to those without BPA exposure (0 BPA/glass), with a significant increase in the total cardiac mast cells (*p* = 0.004) (Figure 7f). Similarly, the mast cell degranulation of the total mast cells (*p* = 0.03) (Figure 7g) and pericardial mast cells (*p* = 0.0001) (Figure 7h) were increased in the 50 BPA group.

### 2.8. Plastic Caging Has No Effect on Mast Cell Numbers or Degranulation in Males during Myocarditis, but Males Have More Degranulating Pericardial Mast Cells Than Females Regardless of Plastic Exposure

While we detected decreased cKit gene expression in the heart of males by qRT-PCR after exposure to plastic caging and BPA in the drinking water (Figure 6), we did not observe a significant change in the total number of mast cells or mast cell degranulation, histologically, with either exposure (Figure 8). However, an important sex difference that we observed in the mast cells histologically was that the exposure to plastic caging and BPA in the drinking water caused a shift in the females from pericardial mast cells that were not degranulating to those that were degranulating (Figure 7c and Figure 9a). However, in males the mast cell numbers/degranulation state did not shift after plastic/BPA exposure, but males had higher numbers of degranulating mast cells regardless of plastic/BPA exposure (Figure 8c; Figure 9b). The data that are shown in Figure 7 and Figure 8 are combined by sex in Figure 9, to illustrate the point more clearly.

### 2.9. Plastic Caging Increases Fibrosis in the Heart during Myocarditis in Females, but Has No Significant Affect in Males

When we examined female BALB/c mice for evidence of fibrosis by determining the collagen levels in the heart by qRT-PCR and fibrosis histologically, we found that plastic caging significantly increased collagen 1 gene expression in the heart (*p* = 0.009) (Figure 10a), and cardiac fibrosis (*p* = 0.023) (Figure 10b). The same result was observed for exposure to 250 μg BPA/L in the drinking water for fibrosis (*p* = 0.0005) (Figure 10b). The one-way ANOVA was significantly different between the groups in females for collagen 1 (*p* = 0.005) (Figure 10a) and for fibrosis (*p* = 0.001) (Figure 10b). In contrast, plastic caging did not significantly alter collagen 1 gene expression (*p* = 0.71) (Figure 10c) or fibrosis in males (*p* = 0.29) (Figure 10d) compared to glass caging; this was a result that was confirmed for BPA exposure in the drinking water (collagen 1 gene expression *p* = 0.71, Figure 10c; and fibrosis *p* = 0.85, Figure 10d). The one-way ANOVA was not significantly different between the groups in males for collagen 1 (*p* = 0.69) (Figure 10c) or for fibrosis (*p* = 0.30) (Figure 10d).

### 2.10. Plastic Caging Decreases ERα and AR and Increases ERβ Expression in the Heart of Males during Myocarditis, but Has No Affect on ERs in Females

The ERs and AR are located on/in immune cells, cardiomyocytes, endothelial cells, and cardiac fibroblasts in both males and females [67,68,69,70]. It is likely that the ratio of ERs-to-ARs on/in immune cells is critical in the sex hormone regulation of inflammation [67,68,69,70]. However, this is a rapidly growing area of research, with many questions that still remain. When comparing plastic caging to glass caging, we found that there was no significant difference in the gene expression of ERs in the heart in females (Figure 11a–c), but they had significantly increased AR expression (*p* = 0.03) (Figure 11d). In contrast, ERβ was significantly increased in females that were exposed to 250 μg BPA/L in their drinking water (*p* = 0.04), compared to 0 BPA/glass (Figure 11b). The one-way ANOVA was not significantly different between the groups in females for ERα (*p* = 0.52) (Figure 11a), ERβ (*p* = 0.02) (Figure 11b), ERRγ (*p* = 0.30) (Figure 11c), or AR (*p* = 0.004) (Figure 11d).

On the other hand, male BALB/c mice who were housed in plastic cages had significantly decreased cardiac gene expression of ERα (*p* = 0.02) (Figure 11e), decreased expression of ERβ (*p* = 0.03) (Figure 11f), and increased expression of the AR (*p* = 0.002) (Figure 11h) in the heart by qRT-PCR. The plastic caging had no significant effect on ERRγ expression in the heart of males with myocarditis compared to the glass cages using qRT-PCR (*p* = 0.09) (Figure 11g). In contrast, 250 μg BPA/L (50 BPA) in the drinking water of males did not significantly alter the expression of ERα (*p* = 0.46), ERβ (*p* = 0.36), ERRγ (*p* = 0.07), or AR (*p* = 0.13) expression in the heart compared to the control water (0 BPA/glass) (Figure 11e–h). The one-way ANOVA was not significantly different between the groups in males for ERα (*p* = 0.02) (Figure 11e), ERβ (*p* = 0.05) (Figure 11f), ERRγ (*p* = 0.08) (Figure 11g), or the AR (*p* = 0.005) (Figure 11h).

### 2.11. Plastic Caging Increases Complement Gene Expression in the Heart of Males with Myocarditis, but Not Females

Because of the important role of the complement pathway in the pathogenesis of CVB3 myocarditis in male mice [71], and because plastic caging and the EPA reference dose of BPA increased CD11b/CR3 expression in the males with myocarditis (*p* = 0.002 and *p* = 0.04, respectively), we examined the expression of complement components and receptors in the heart of mice who were housed in plastic caging or exposed to 250 μg BPA/L in their drinking water. We found that the plastic caging had no significant effect on the complement components in females with plastic caging (Table 1). In contrast, the plastic caging significantly increased the expression of CD11b/CR3 (*p* = 0.002) (Figure 12a), C4b (*p* = 0.002) (Figure 12c), and the mast cell anaphylatoxin complement receptor C5aR1 (*p* = 0.006) (Figure 12d) in males during acute myocarditis compared to glass caging. These complement components were also significantly increased in males after exposure to BPA in their drinking water (Figure 12a,b,d). One-way ANOVA was significantly different between the groups in females for CR3 (*p* = 0.004) (Figure 12a), C3aR1 (*p* = 0.17) (Figure 12b), C4b (*p* = 0.004) (Figure 12c), and C5aR1 (*p* = 0.008) (Figure 12d).

## 3. Discussion

In this study, we found that plastic caging alone increased myocarditis and fibrosis in adult male and female BALB/c mice, respectively, with a similar result observed after a high-dose exposure of BPA in their drinking water. These findings suggest that BPA that ‘leaches’ from plastic water bottles, and possibly also from plastic cages that house the mice, is able to alter viral myocarditis in a sex-specific manner. Previously, it has been reported that BPA can leach from polysulfone and polycarbonate cages that have been exposed to high temperatures, as occurs during the autoclave sterilization process [72]. However, in glass and polypropylene cages, the leakage of BPA was not observed [72]. BPA was also found to be released from hemodialyzers that were comprised of polysulfone and polycarbonate plastic that had undergone heat disinfection [73]. BPA is known to leach from plastic into water, such as water bottles that are used by people and in caging that is used to house research animals. Studies have determined that the exposure route of BPA influences its pharmacokinetics and the clinical relevance of the animal studies. The oral exposure of BPA has been found to more closely match the levels that are found in humans, compared to subcutaneous injection or bolus gavage routes [74]. Future studies will need to determine whether BPA leaching from plastic water bottles or plastic cages is sufficient to increase myocarditis in mice, or if it requires both sources.

Male BALB/c mice in our model of CVB3 myocarditis typically have inflammation ranging from 30 to 50% of the heart section [22,25]; these are data that have been obtained with traditional plastic caging and bedding. Our findings here suggest that the traditional plastic caging that is used to house mice may contribute to the sex differences in viral myocarditis, particularly an increase in the complement components on CD11b+ macrophages, neutrophils, and mast cells. Interestingly, BPA has also been found to significantly increase the number of CD11b+ microglia (which are brain macrophages) in the brains of male rats, but not in females [61]. Importantly, the same sex difference that we observe for myocarditis in BALB/c mice who are housed in plastic cages (i.e., worse in males) exists for myocarditis patients, [11,14,19] suggesting that exposure to plastics in our environment may promote sex differences in inflammation in the heart in response to viral infection in humans. To our knowledge, no studies have been conducted to determine the BPA levels in the blood or urine of patients with myocarditis. BPA exposure has been found to activate macrophages to promote a proinflammatory response including elevated TLR4, and the proinflammatory and profibrotic cytokines TNFα, IL-1β, IL-6, and IL-8, within lung tissue from 4-week-old female C57BL/6 mice [75,76,77,78]. Importantly, the sex differences in myocarditis after plastic caging in this study were not due to alterations in viral replication based on VP1 gene expression. Previously, we described that viral replication in the heart, using a plaque assay, does not differ by sex during acute CVB3 myocarditis, with both sexes clearing the virus by day 14 [25,71,79].

Complement cascade genes are known to be important in promoting human [80] and CVB3 myocarditis in mice [28]. The upregulation of CD11b/CR3 and C5aR1 in male mice due to plastic caging (and all complement components with high-dose BPA exposure) suggests mast cell activation, yet the cKit levels were significantly decreased in the hearts of males by both plastic caging and BPA exposure, and there was no increase in mast cell number or degranulation histologically for either exposure route. Perhaps there was not an increase because the mast cell numbers and pericardial mast cell degranulation were already high in males prior to BPA exposure. In contrast, the primary immune effect of BPA exposure on female BALB/c mice was to increase mast cell numbers, cKit expression, and pericardial mast cell degranulation and fibrosis. Previously, we reported that 25 μg/L BPA in the drinking water of the female BALB/c mice was able to increase myocarditis by activating mast cells [59]. Our data here suggest that BPA exposure from caging increases cardiac inflammation after CVB3 infection by different immunologic mechanisms in males and females.

BPA is known to be an endocrine disruptor particularly targeting the ER, due to its hormone-like properties [59]. As a result, its binding to ARs by competition with 5*a-*dihydrotestosterone (DHT) allows it to disrupt the function of the reproductive pathways in males. Interestingly, a cross-sectional study found that BPA exposure in men led to higher testosterone levels in the sera [81,82]. We have previously shown that testosterone increases myocardial inflammation in male mice and humans with myocarditis, specifically increasing CD11b/CR3+ immune cells [22,26,83]. If BPA leads to increased testosterone levels, this could explain, at least in part, the ability of BPA to increase myocarditis in males. Studies have found that BPA can bind to the AR [62], which could then directly activate the AR, leading to increased myocarditis in males. However, the AR expression in the heart was significantly decreased with plastic caging compared to glass caging, but it can be difficult to interpret hormone receptor function using gene expression, which may decrease due to the activation or engagement of the receptor [84]. Importantly, BPA has been found to alter the metabolism of genes that are involved in hormone metabolism, including TSPO/STAR, which are responsible for cholesterol transport into the mitochondria and the production of sex steroids by macrophages and other cells [85,86,87]. Previously, we have shown that TSPO is expressed primarily in CD11b+ immune cells in the hearts of men and male mice with myocarditis [22,84,88]. In this study, we observed that plastic caging caused significate alterations in the hormone receptor expression in the heart of male mice. BPA has been found to act through ERβ to increase cardiac arrhythmias and other cardiac complications in animal models [44,89]. In CVB3 myocarditis, ERβ signaling was found to promote myocarditis in male and female mice who were treated with the ERβ agonist diarylpropionitrile [24,90]. We also previously showed that female mice given 25 μg/L BPA in their drinking water had elevated ERβ levels in the heart and increased myocarditis [59], which was also increased by 250 μg/L BPA in the drinking water in this study. Additionally, here we found that males housed in plastic caging had significantly increased ERβ expression, which may promote myocardial inflammation. In contrast, ERα is believed to mediate most of the cardioprotective effects of estrogen in women and female mice [67]. Estrogen (17β-estradiol), via ERα, has been found to reduce myocardial inflammation during CVB3 myocarditis [23,70,90,91], which was demonstrated by the elevated cardiac inflammation in ERα knockout mice, while CVB3-infected male mice treated with the ERα agonist propyl pyrazole triol had reduced inflammation [24,90]. Fibrosis does not typically occur during acute CVB3 myocarditis, but we previously found that BPA increased fibrosis in the hearts of female mice given 25 μg/L BPA in their drinking water [59]. Cardiac fibrosis and collagen type I expression were also increased in female mice in this study after exposure to 250 μg/L BPA in their drinking water. In a separate study, we previously showed that α1-antichymotrypsin (Serpin A3n), which is released exclusively from mast cells, is upregulated during CVB3 myocarditis and promotes cardiac remodeling and fibrosis [22] suggesting that mast cell degranulation, caused by plastic caging in this study, increased fibrosis in the females. The lack of mast cell degranulation and fibrosis in males exposed to plastic, even though they had elevated macrophages and neutrophils, further highlights the important role of mast cells in promoting remodeling and fibrosis during myocarditis.

Recent articles have begun to address concerns associated with animal studies that are attempting to understand the role of endocrine disruptors, such as BPA, on immune function and/or disease. These articles have brought up a number of potentially confounding issues, including plastic cages, food and bedding containing the phytoestrogen soy or other phytoestrogens, enormous variation in the doses of BPA used, and varying exposure methods [92]. Our findings suggest that traditional plastic caging may affect the immune response to viral infection in a sex-specific manner, at least in BALB/c mice. Future studies will need to determine whether our findings are specific to viral myocarditis or require viral infection, or if BPA that leaches from plastic water bottles and/or cages can alter the normal physiology of animals or cells in a sex-specific manner. If so, endocrine disruptors could be affecting not only viral myocarditis, but also many of the experiments that are conducted by basic researchers.

BPA and other endocrine disruptors that leach out of plastic are ubiquitous in our environment. Increasing numbers of clinical and basic science studies report an association between BPA levels and worse cardiovascular disease and outcomes. Our findings suggest that BPA exposure in the environment may promote sex differences in cardiac inflammation following viral infection. Future studies are needed to determine whether these chemical endocrine disruptors contribute to worse myocarditis outcomes in men or women with common viral infections, such as coxsackievirus, influenza, and SARS-CoV-2.

## 4. Materials and Methods

### 4.1. Animal Care Ethics Statement

Mice were used in strict accordance with the recommendations in the Guide for the Care and Use the Laboratory Animals of the National Institutes of Health. Mice were maintained under pathogen-free conditions in the animal facility at the Johns Hopkins School of Medicine and at Mayo Clinic Florida, and approval was obtained from the Animal Care and Use Committee at Johns Hopkins University and Mayo Clinic Florida for all procedures (IACUC numbers (Approval date): A30315 (27 October 2015), A00003983 (3 January 2019)). Mice were sacrificed according to the Guide for the Care and Use of Laboratory Animals of the National Institutes of Health.

### 4.2. CVB3-Induced Myocarditis Model

Male and female BALB/c (stock #651) 6–8 week old adult mice were obtained from the Jackson Laboratory (Bar Harbor, ME). Mice were maintained under pathogen-free conditions in the animal facility at the Johns Hopkins School of Medicine or the Mayo Clinic Florida animal facility. Generally, 10 mice per group/sex were used for all experiments, unless otherwise indicated. Mice were placed in plastic cages with plastic water bottles, or glass caging with glass water bottles. When the mice were 8 weeks old, they were inoculated intraperitoneally (i.p.) with sterile phosphate-buffered saline (PBS) or 10^3^ plaque forming units (PFU) of heart-passaged stock of coxsackie virus B3 (CVB3) on day 0, and acute myocarditis examined at day 10 pi, as previously described [93,94]. CVB3 (i.e., Nancy strain) was originally obtained from the American Type Culture Collection (ATCC; Manassas, VA, USA) and grown in Vero cells (ATCC), as previously described [94].

### 4.3. Bisphenol A and Bedding

The dose of BPA (Sigma, St. Louis, MO, USA) that was administered was 250 µg BPA/L in drinking water, which is equivalent to an estimated intake of 50 µg BPA/kg body weight (BW), based on predicted exposure levels in the human population [65]. At the time of the development of this project, Jenkins et al. was the only study available that assessed the effect of BPA in a mouse model using oral exposure in drinking water. The EPA reference dose was calculated using a safety factor of 1000×, the lowest observable adverse effect level (LOAEL) [95]. The EPA reference dose is defined as an estimate of the daily exposure to a susceptible individual without an appreciable risk of deleterious effects during a lifetime. Estimated intake of BPA for mice in drinking water was based on [65]. They found that a 20 g mouse drinks approximately 4 mL of water a day and reported that BPA is stable for one week in water [65,96]. For this reason, as well as to provide the mice with fresh water, control and BPA water were replaced each week of the experiment.

All experiments used bedding (Envigo-Tekland, 7990.BG, Minneapolis, MN, USA) and food (Envigo-Tekland, 2020X) from Envigo (Minneapolis, MN, USA) that was free of soy and phytoestrogen to exclude other naturally occurring endocrine disruptors. BPA was given to mice dissolved in drinking water for two weeks prior to inoculation i.p. with 10^3^ PFU of heart-passaged stock of CVB3 on day 0 and acute myocarditis examined at day 10 pi, as previously described [22,94]. BPA exposure was continued from day 0 of viral infection until harvest at day 10 pi. At harvest, heart tissue was divided in half and each half of the heart was used for histology, IHC or qRT-PCR.

### 4.4. Histology

Mouse hearts were cut longitudinally and fixed in 10% phosphate-buffered formalin and embedded in paraffin for histological analysis. Five micron sections were stained with hematoxylin and eosin (H&E) to detect inflammation, picrosirius red to detect collagen or toluidine blue to detect mast cell granules. Myocarditis and fibrosis were assessed as the percentage of the heart with inflammation or fibrosis compared to the overall size of the heart section using a microscope eyepiece grid, as previously described [22,83,88,97]. Sections were scored by at least two individuals blinded to the treatment group.

### 4.5. Quantitative Real-Time PCR

#### 4.5.1. RNA Isolation of Heart Tissue

At harvest, half of the heart was collected and stored at −80 °C for RNA isolation. Hearts were homogenized and lysed using Tissuelyser (Qiagen, Germantown, MD, USA), with 7 mm stainless steel beads in RTL buffer with 0.5% DX buffer to reduce foam. The homogenate was then placed in an automated RNA isolation and purification instrument, QIAcube, with reagents for RNase easy fibrous mini kit including a DNase and proteinase K step (Qiagen, Germantown, MD, USA). RNA was eluted into 30 μL. If the heart had been divided in the earlier step, the eluted RNA was pooled prior to being aliquoted. RNA quantification was determined in μg/μL using NanoDrop (Thermo Scientific, Waltham, MA, USA).

#### 4.5.2. qRT-PCR Method

Total RNA from mouse hearts was assessed by quantitative real-time (qRT) PCR using assay-on-demand primers and probe sets and the ABI 7000 Taqman system from Applied Biosystems (Foster City, CA, USA) after RNA was converted to cDNA using high-capacity cDNA reverse transcriptase kit (Applied Biosystems, Foster City, CA, USA), as previously described [84,98]. Data are shown as relative gene expression (RGE) normalized to the housekeeping gene hypoxanthine phosphoribosyltransferase 1 (Hprt). All the following primers listed were purchased from Thermo-Scientific (Waltham, MA, USA): HPRT (Mm03024075_m1), CD45 (Mm00448522_m1), CD11b (Mm00434455_m1), F480 (Mm00802529_m1), GR1 (Mm00439154_m1), CD14 (Mm00438094_g1), CD3e (Mm01179194), CD4 (Mm00442754_m1), CD8a (Mm01182107_g1), cKit (Mm00445212_m1), C3aR1 (Mm02620006_s1), C4b (Mm00437893_g1), C5aR1 (Mm00500292_s1), Esr1 (Mm00433149_m1), Esr2 (Mm00599821_m1), Esrrg (Mm01314576_m1), AR (Mm00442688_m1), Col1a1 (Mm00801666_g1). Gene expression was analyzed by assessing comparative quantification, which utilizes cycle threshold (Ct) for each primer to calculate the delta Ct (∆Ct), which is the threshold cycle comparison between the gene of interest and the housekeeping gene. This is then used to calculate the relative gene expression using the formula RGE = 2 − (∆Ct − ∆Ct(max)).

#### 4.5.3. Measurement of CVB3 Genome VP1 Levels by qRT-PCR

Probe sets to detect CVB3 VP1 were developed by Antoniak et al. and obtained from Integrated DNA Technologies (Coralville, IA, USA) [66]. Probe sets are as follows: CVB3 forward, 5′-CCCTGAATGCGGCTAATCC-3′; CVB3 reverse, 5′-ATTGTCACCATAAGCAGCCA-3′; CVB3 probe, 5′-FAM-TGCAGCGGAACCG-TAMRA-3′.

### 4.6. Immunohistochemistry

Five micron sections of the heart were stained with CD3 (Abcam, ab16669, 1:200, rabbit), CD11b (Abcam, Cambridge, United Kingdom, ab133357, 1:3000, rabbit), CD45 (Biolegend, San Diego, CA, USA, 103102, 1:200, rat), F4/80 (BioRad, Hercules, CA, USA, MCA497G, 1:250, rat), or GR-1 (Invitrogen, Waltham, MA, USA14-5931-85, 1:150, rat). Secondary antibody for rabbit utilized Envision+ anti-rabbit labeled polymer (K4006) and rat-on-rodent kit (RT517) (Biocare, Pacheco, CA, USA) for rat antibodies. Stained slides were scanned using an Aperio AT2 slide scanner (Leica, Wetzlar, Germany). Ventricles of cardiac sections were manually selected by a lab member blinded to the study groups for analysis. The default “positive pixel” algorithm was modified for each stain by adjusting the hue width parameter so that the program’s selection of positive and negative pixel counted accurately reflected each stain. The hue value for all algorithms used was 0.1 (brown). The ventricles of each heart were analyzed using Aperio eSlide Manager (Leica, Wetzlar, Germany) with the aforementioned algorithms. Stain positivity (% Positive) for CD3, CD11b, CD45, and F4/80 were determined with the default “Positivity” parameter (positivity = number of positive pixels/(number of positive + number of negative pixels)). For GR1 slides, which had fewer cells that stained positive than the other markers, positivity was determined using the default “Nsr” output (Nsr = number of strongly positive pixels/(number of weakly positive + number of positive + number of strongly positive pixels)) where “strongly positive pixels” indicated intensity of the stain.

### 4.7. Statistical Analysis

The normally distributed data comparing two groups were analyzed using a two-tailed Student’s *t-*test. The data comparing three groups were analyzed using one-way ANOVA with Holm–Šídák’s multiple comparisons test. The data comparing two parameters (mast cell degranulation status and experimental groups) was analyzed using two-way ANOVA with Holm–Šídák’s multiple comparisons test. The data are expressed as scatter plot and mean ± SEM. A value of *p* < 0.05 was considered significant. Statistical analysis was performed in GraphPad Prism 9.0.2.

## Figures and Tables

**Figure 1 ijms-22-08834-f001:**
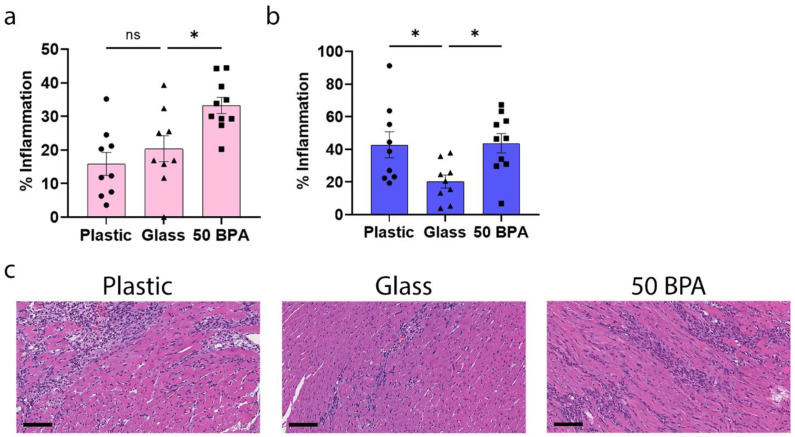
Plastic caging increases myocardial inflammation during acute myocarditis in male BALB/c mice. (**a**) Female (pink) and (**b**) male (blue) BALB/c mice were given normal drinking water (drinking water that did not contain BPA) for 2 weeks and housed either in glass cages/water bottles (glass) or plastic cages/water bottles (plastic) with soy-free food and bedding, or BALB/c mice housed in glass cages/water bottles with soy-free food and bedding were either given normal drinking water (drinking water that did not contain BPA) (glass) or 250 μg BPA/L water (50 BPA) for 2 weeks. After BPA exposure, mice received an intraperitoneal (i.p.) injection with 10^3^ plaque-forming units (PFU) of CVB3 on day 0 and harvested at day 10 post infection (pi). Myocarditis was assessed as % inflammation compared to the total size of the heart section with H&E using a microscope grid. Data shown as scatter plot and mean +/−SEM using one-way ANOVA with Holm–Šídák’s multiple comparisons test with 9–10 mice/group (* *p* < 0.05). (**c**) Representative H&E images of cardiac inflammation in plastic caging (plastic), glass caging (glass) or with 50 BPA in drinking water (50 BPA) (magnification 200×, scale bar 100 μm).

**Figure 2 ijms-22-08834-f002:**
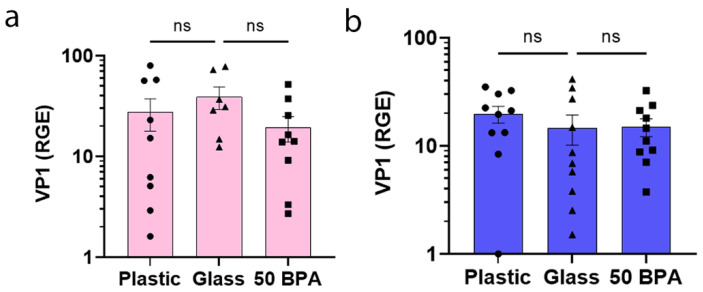
Plastic caging or BPA did not increase cardiac VP1 viral gene expression during acute myocarditis. (**a**) Female (pink) and (**b**) male (blue) adult BALB/c mice were given normal drinking water (drinking water that did not contain BPA) for 2 weeks and housed in glass cages/water bottles (glass) or plastic cages/water bottles (plastic) caging with soy-free food and bedding. BALB/c mice housed in glass cages/water bottles with soy-free food and bedding received either normal drinking water (drinking water that did not contain BPA) (glass) or 250 μg BPA/L water (50 BPA) for 2 weeks. Mice were injected i.p. with 10^3^ PFU of CVB3 on day 0 and harvested at day 10 pi. Relative gene expression (RGE) of CVB3 VP1 levels in the heart were determined using qRT-PCR compared to the housekeeping gene hypoxanthine phosphoribosyltransferase (Hprt). Data shown as scatter plot and mean +/−SEM using one-way ANOVA with Holm–Šídák’s multiple comparisons test with 7–10 mice/group.

**Figure 3 ijms-22-08834-f003:**
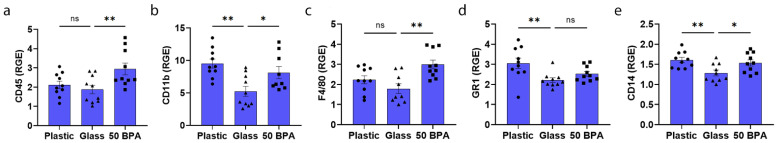
Plastic caging increases macrophage and neutrophil markers during acute myocarditis in BALB/c males. Male BALB/c mice were given normal drinking water (drinking water that did not contain BPA) for 2 weeks and housed either in glass cages/water bottles (glass) or plastic cages/water bottles (plastic) with soy-free food and bedding. BALB/c mice were housed in glass cages/water bottles with soy-free food and bedding and administered either normal drinking water (drinking water that did not contain BPA) (glass) or 250 μg BPA/L water (50 BPA) for 2 weeks. Mice were injected i.p. with 10^3^ PFU of CVB3 on day 0 and harvested at day 10 pi. Relative gene expression (RGE) of whole hearts by qRT-PCR was used to assess (**a**) CD45 (total lymphocytes), (**b**) CD11b+ cells (i.e., macrophages, neutrophils, mast cells), (**c**) F4/80+ macrophages, (**d**) GR1 (neutrophils), (**e**) CD14 (part of the TLR4 signaling complex), compared to the housekeeping gene Hprt. Data shown as scatter plot and mean +/−SEM using one-way ANOVA with Holm–Šídák’s multiple comparisons test with 9–10 mice/group (* *p* < 0.05), (** *p* < 0.01).

**Figure 4 ijms-22-08834-f004:**
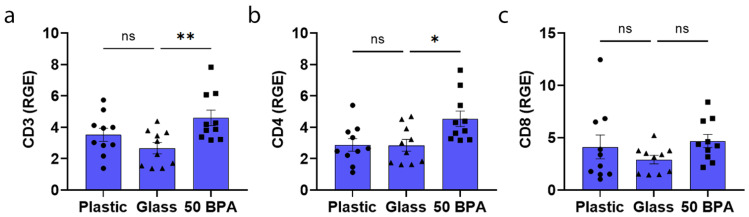
Plastic caging has no effect on expression of T-cell markers during acute myocarditis in BALB/c males. Male BALB/c mice were given normal drinking water (drinking water that did not contain BPA) for 2 weeks and housed either in glass cages/water bottles (glass) or plastic cages/water bottles (plastic) with soy-free food and bedding. BALB/c mice housed in glass cages/water bottles with soy-free food and bedding were given either normal drinking water (drinking water that did not contain BPA) (glass) or 250 μg BPA/L (50 BPA) in drinking water for 2 weeks. Mice were injected i.p. with 10^3^ PFU of CVB3 on day 0 and harvested at day 10 pi. Relative gene expression (RGE) was used in whole hearts by qRT-PCR to assess (**a**) CD3+ T cells, (**b**) CD4+ T cells and (**c**) CD8+ T cells compared to the housekeeping gene Hprt. Data shown as scatter plot and mean +/−SEM using one-way ANOVA with Holm–Šídák’s multiple comparisons test with 10 mice/group (* *p* < 0.05), (** *p* < 0.01).

**Figure 5 ijms-22-08834-f005:**
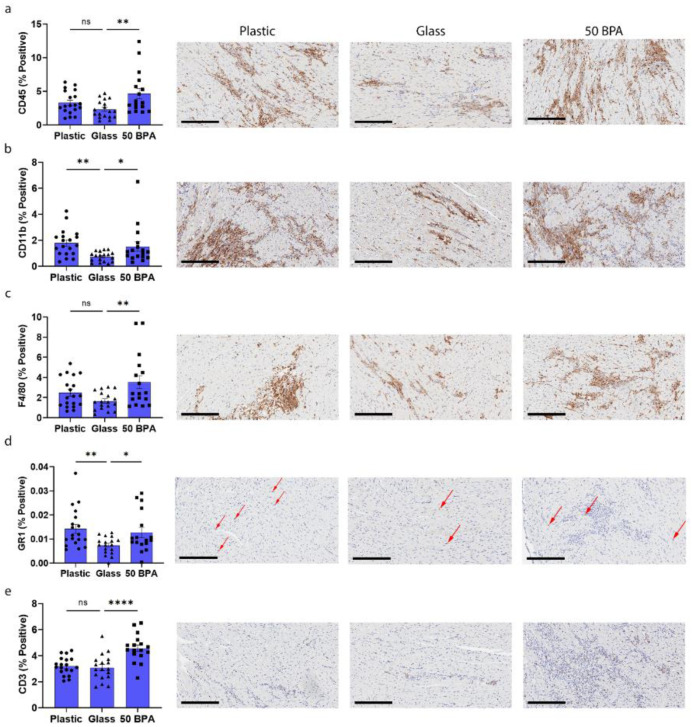
Plastic caging increases macrophages and neutrophils during acute myocarditis in BALB/c males. Male BALB/c mice were given normal drinking water (drinking water that did not contain BPA) for 2 weeks and housed either in glass cages/water bottles (glass) or plastic cages/water bottles (plastic) with soy-free food and bedding. BALB/c mice were housed in glass cages/water bottles with soy-free food and bedding and administered either normal drinking water (drinking water that did not contain BPA) (glass) or 250 μg BPA/L water (50 BPA) for 2 weeks. Mice were injected i.p. with 10^3^ PFU of CVB3 on day 0 and harvested at day 10 pi. Five micron sections were stained with antibodies against (**a**) CD45 (total lymphocytes), (**b**) CD11b+ cells (i.e., macrophages, neutrophils, mast cells), (**c**) F4/80+ macrophages, (**d**) GR1 (neutrophils), (**e**) CD3+ T cells, and positive pixels were compared to total positive and negative pixels to determine % positive. Data shown as scatter plot and mean +/−SEM using one-way ANOVA with Holm–Šídák’s multiple comparisons test for 17–19 mice/group (* *p* < 0.05), (** *p* < 0.01), (**** *p* < 0.0001). Representative images of positive-stained cells (brown) in plastic caging (plastic), glass caging (glass) and 50 BPA in drinking water (50 BPA) in glass cages shown for (**a**) CD45, (**b**) CD11b, (**c**) F4/80, (**d**) GR1, and (**e**) CD3. Red arrow points to brown GR1-positive cells (magnification 200×, scale bar 200 μm).

**Figure 6 ijms-22-08834-f006:**
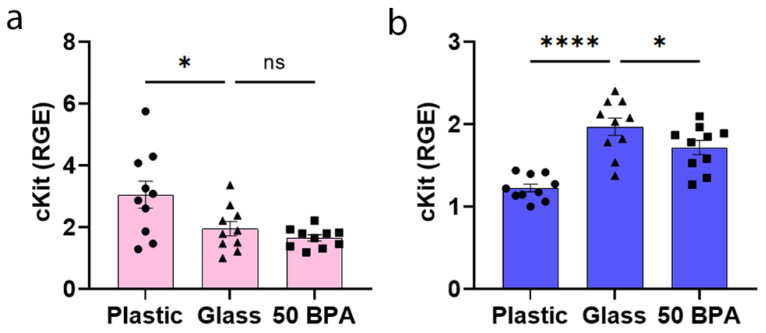
Plastic caging increases the mast cell marker cKit during acute myocarditis in BALB/c females, but decreases its expression in males. (**a**) Female (pink) and (**b**) male (blue) BALB/c mice were given normal drinking water (drinking water that did not contain BPA) for 2 weeks and housed either in glass cages/water bottles (glass) or plastic cages/water bottles (plastic) with soy-free food and bedding. BALB/c mice housed in glass cages/water bottles with soy-free food and bedding were given either normal drinking water (drinking water that did not contain BPA) (glass) or 250 μg BPA/L water (50 BPA) for 2 weeks. Mice were injected i.p. with 10^3^ PFU CVB3 on day 0 and harvested at day 10 pi. The cKit levels in the heart were determined by qRT-PCR for relative gene expression (RGE) compared to the housekeeping gene Hprt. Data shown as scatter plot and mean +/−SEM using one-way ANOVA with Holm–Šídák’s multiple comparisons test with 10 mice/group (* *p* < 0.05), (**** *p* < 0.0001).

**Figure 7 ijms-22-08834-f007:**
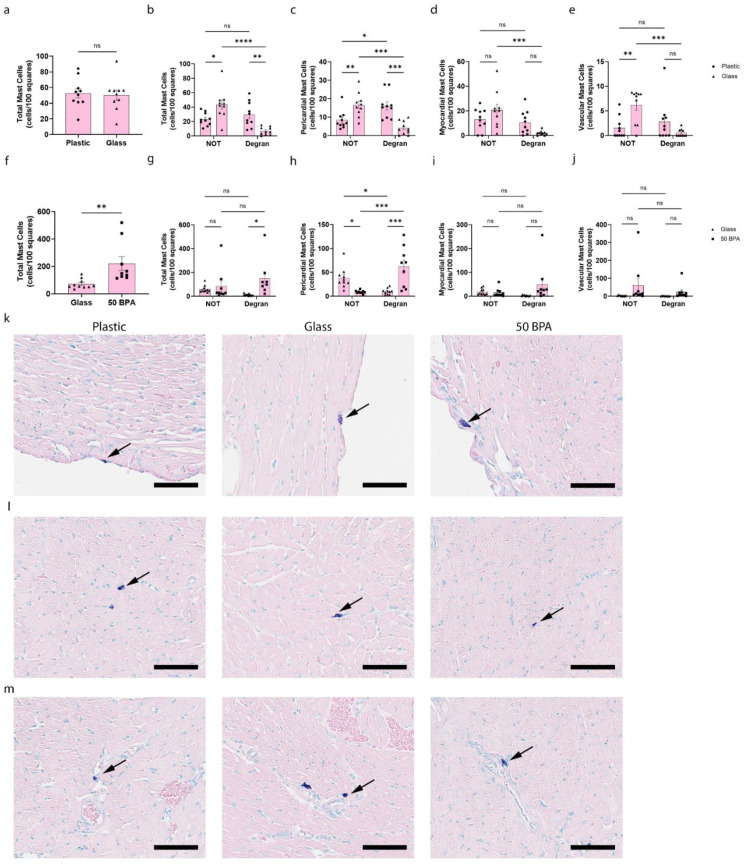
Plastic caging increases pericardial mast cell numbers and degranulation in females during myocarditis. Female BALB/c mice (pink) were given (**a**–**e**) normal drinking water (drinking water that did not contain BPA) for 2 weeks and housed either in glass cages/water bottles (glass) or plastic cages/water bottles (plastic) with soy-free food and bedding. (**f**–**j**) BALB/c mice housed in glass cages/water bottles with soy-free food and bedding were given either normal drinking water (drinking water that did not contain BPA) (glass) or 250 μg BPA/L water (50 BPA) for 2 weeks. Mice were injected i.p. with 10^3^ PFU of CVB3 on day 0 and harvested at day 10 pi. Mast cell scoring was completed using the 40× high-power objective on a compound microscope. Mast cells were counted and categorized according to whether they were degranulated or not and by their location as pericardial, myocardial or near vessels. Data shown as scatter plot and mean +/−SEM using a Student’s *t*-test with 9–10 mice/group (* *p* < 0.05), (** *p* < 0.01), (*** *p* < 0.001), (**** *p* < 0.0001). Representative histology images of mast cells (purple, arrow). Representative images in plastic caging (plastic), glass caging (glass), or 50 BPA in drinking water (50 BPA) in glass cages show mast cells associated with the (**k**) pericardium, (**l**) myocardium or (**m**) vessels (magnification 400×, scale bar 60 μm).

**Figure 8 ijms-22-08834-f008:**
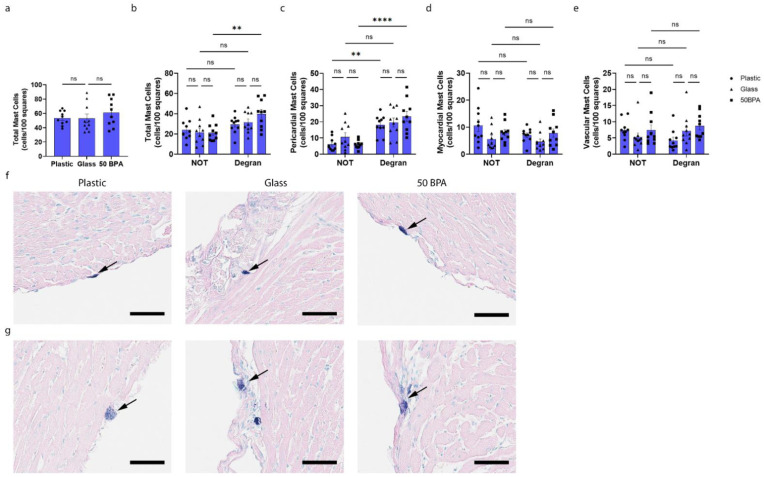
Plastic caging does not alter mast cell numbers or degranulation in males during myocarditis. Male BALB/c mice (blue) were given normal drinking water (drinking water that did not contain BPA) for 2 weeks and housed either in glass cages/water bottles (glass) or plastic cages/water bottles (plastic) with soy-free food and bedding. BALB/c mice housed in glass cages/water bottles with soy-free food and bedding were given either normal drinking water (drinking water that did not contain BPA) (0 BPA/glass) or 250 μg BPA/L water (50 BPA) for 2 weeks. Mice were injected i.p. with 10^3^ PFU of CVB3 on day 0 and harvested at day 10 pi. Mast cell scoring was completed using the 40× high-power objective on a compound microscope. (**a**–**e**) Mast cells were counted and categorized according to whether they were degranulated (small blue granules can be observed released from the cell at high power) or not and by their location as pericardial, myocardial or near vessels. Data shown as scatter plot and mean +/−SEM using one-way ANOVA with Holm–Šídák’s multiple comparisons test with 10 mice/group (** *p* < 0.01), (**** *p* < 0.0001). (**f**,**g**) Representative histology images of mast cells (purple, arrow). Representative images in plastic caging (plastic), glass caging (glass) or with 50 BPA in drinking water in glass cages (50 BPA) show (**f**) mast cells that are not degranulating (NOT) versus (**g**) degranulating mast cells (degran) with identifiable mast cell granules outside of the cell (magnification 400×, scale bar 60 μm).

**Figure 9 ijms-22-08834-f009:**
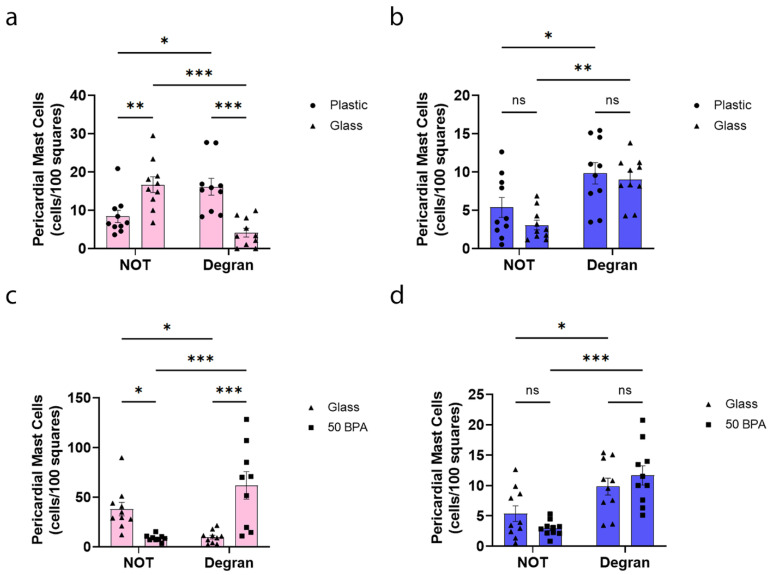
Plastic caging associated with pericardial mast cell degranulation in females, but not in males, where mast cells degranulate regardless of plastic exposure. (**a**,**c**) Female (pink) and (**b**,**d**) male (blue) BALB/c mice were given (**a**,**b**) normal drinking water (drinking water that did not contain BPA) for 2 weeks and housed either in glass cages/water bottles (glass) or plastic cages/water bottles (plastic) with soy-free food and bedding. (**c**,**d**) BALB/c mice housed in glass cages/water bottles with soy-free food and bedding were given either normal drinking water (drinking water that did not contain BPA) (glass) or 250 μg BPA/L water (50 BPA) in glass cages for 2 weeks. Mice were injected i.p. with 10^3^ PFU of CVB3 on day 0 and harvested at day 10 pi. Mast cell scoring was completed using the 40× high-power objective on a compound microscope. Mast cells were counted and categorized according to whether they were degranulated or not and by their location as pericardial. Data shown as scatter plot and mean +/−SEM using a Student’s *t*-test with 9–10 mice/group (* *p* < 0.05), (** *p* < 0.01), (*** *p* < 0.001).

**Figure 10 ijms-22-08834-f010:**
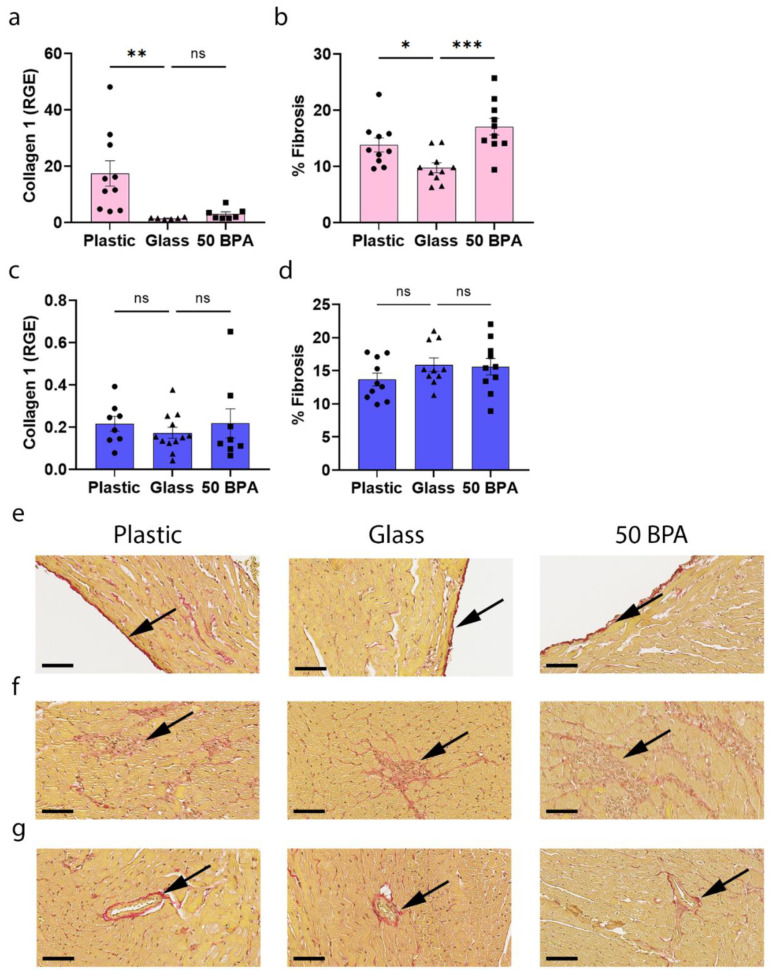
Plastic caging increases collagen gene expression and fibrosis in the heart of females, but not male mice during myocarditis. (**a**,**b**) Female (pink) and (**c**,**d**) male (blue) BALB/c mice were given normal drinking water (drinking water that did not contain BPA) for 2 weeks and housed either in glass cages/water bottles (glass) or plastic cages/water bottles (plastic) with soy-free food and bedding. BALB/c mice housed in glass caging with soy-free food and bedding were given either normal drinking water (drinking water that did not contain BPA) (0 BPA/glass) for 2 weeks or 250 μg BPA/L water (50 BPA). Mice were injected i.p. with 10^3^ PFU of CVB3 i.p. on day 0 and harvested at day 10 pi. (**a**,**c**) Collagen 1 levels in the heart were measured using qRT-PCR to determine relative gene expression (RGE) of collagen I compared to the housekeeping gene Hprt. (**b**,**d**) Fibrosis was assessed as the % fibrosis compared to the total size of the heart section using Sirius Red-stained sections and a microscope eyepiece grid. Data shown as scatter plot and mean +/−SEM using one-way ANOVA with Holm–Šídák’s multiple comparisons test with 8–12 mice/group (* *p* < 0.05), (** *p* < 0.01), (*** *p* < 0.001). Representative images of fibrosis detected with Sirius Red (arrow) shown in the (**e**) pericardium, (**f**) myocardium or (**g**) vessels (magnification 400×, scale bar 60 μm).

**Figure 11 ijms-22-08834-f011:**
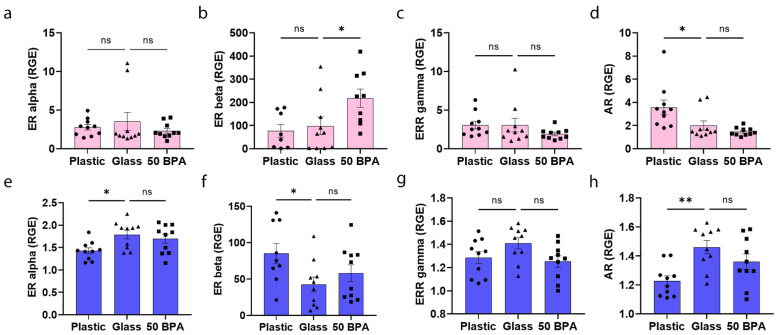
Plastic caging decreases ERα and AR and increases ERβ expression in the heart of males, but has no effect on ERs in females during myocarditis. (**a**–**d**) Female (pink) and (**e**–**h**) male (blue) BALB/c mice were given normal drinking water (drinking water that did not contain BPA) for 2 weeks and housed either in glass cages/water bottles (glass) or plastic cages/water bottles (plastic) with soy-free food and bedding. BALB/c mice housed in glass cages/water bottles with soy-free food and bedding were given either normal drinking water (drinking water that did not contain BPA) (0 BPA/glass) or 250 μg BPA/L water (50 BPA) for 2 weeks. Mice were injected i.p. with 10^3^ PFU of CVB3 on day 0 and harvested at day 10 pi. Relative gene expression (RGE) was assessed in whole hearts by qRT-PCR. (* *p* < 0.05), (** *p* < 0.01). Expression of (**a**,**e**) ERα, (**b**,**f**) ERβ, (**c**,**g**) ERRγ, and (**d**,**h**) AR were assessed compared to the housekeeping gene Hprt. Data shown as scatter plot and mean +/−SEM using one-way ANOVA with Holm–Šídák’s multiple comparisons test with 8–10 mice/group.

**Figure 12 ijms-22-08834-f012:**
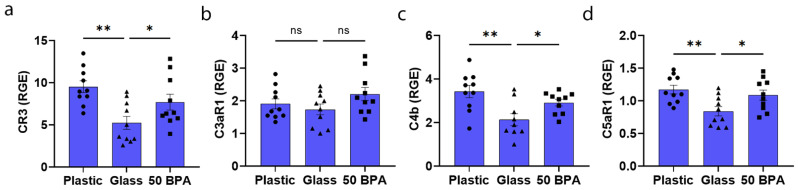
Plastic caging increases complement markers in males with myocarditis, but not in females**.** Male (blue) BALB/c mice were given normal drinking water (drinking water that did not contain BPA) for 2 weeks and housed either in glass cages/water bottles (glass) or plastic cages/water bottles (plastic) with soy-free food and bedding. BALB/c mice housed in glass cages/water bottles with soy-free food and bedding were given either normal drinking water (drinking water that did not contain BPA) (0 BPA/glass) or 250 μg BPA/L water (50 BPA) for 2 weeks. Mice were injected i.p. with 10^3^ PFU of CVB3 on day 0 and harvested at day 10 pi. Relative gene expression (RGE) was used in whole hearts by qRT-PCR to assess (**a**) complement receptor 3 (CR3), (**b**) complement component 3 antagonist receptor 1 (C3aR1), (**c**) complement component C4b, or (**d**) complement component 5 antagonist receptor 1 (C5aR1), compared to the housekeeping gene Hprt. Data shown as scatter plot and mean +/−SEM using one-way ANOVA with Holm–Šídák’s multiple comparisons test with 10 mice/group (* *p* < 0.05), (** *p* < 0.01).

**Table 1 ijms-22-08834-t001:** Effect of plastic vs. glass caging on female mice on expression of complement and fibro-inflammatory markers in the heart using qRT-PCR.

Cell Marker	Description	Glass	Plastic	*p*-Value
Complement				
CR3	complement receptor 3	6.2 ± 1.1	5.5 ± 1.3	0.34
C3aR1	complement component 3a receptor 1	3.5 ± 0.6	4.7 ± 0.8	0.13
C4b	complement component 4b	4.6 ± 0.7	6.0 ± 1.2	0.15
C5aR1	complement component 5a receptor 1	3.3 ± 0.5	3.4 ± 0.5	0.46
Fibro-inflammatory IL-1R2	interleukin 1 receptor, type II	1.9 ± 0.2	2.4 ± 0.4	0.08
Timp-1	tissue inhibitor matrix metalloproteinase 1	7.5 ± 1.6	9.6 ± 2.3	0.23

## Data Availability

The data presented in this study are openly available in FigShare at doi:10.6084/m9.figshare.15176019.

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
