# Peer review of "Sex-Specific Effects of Plastic Caging in Murine Viral Myocarditis"

_ijms, 2021, doi:10.3390/ijms22168834_

Round 1

Reviewer 1 Report

The manuscript by Bruno et al sought to investigate the effect of plastic caging or high level of BPA on acute myocarditis in male and female mice. They found that exposure to plastic caging increased complement activation and the numbers of macrophages and neutrophils in male mice which worsened myocarditis, whereas female mice had elevated mast cell activation and fibrosis. There are many significant deficits.

  1. The exact same data of CD45, CD11b, and F4/80 in Table 2 was published in their previous publication (Table 3 Bruno KA et al. Front Endocrinol (Lausanne). 2019. doi:10.3389/fendo.2019.00598).
  2. Did the authors measure the equivalent BPA level in the plastic bottle/cage? What would the equivalent BPA level be? Since the water was changed every week, did the BPA level increase in the plastic bottle/cage over time? Where the BPA used in the glass bottle was obtained?
  3. Was there any change in cardiac function after the development of myocarditis? Did BPA or plastic caging worsen cardiac function?
  4. Figure 1 e-g was not mentioned in the text. It is better to label them.
  5. It seemed that the “Glass” and “0 BPA” groups in each figure are identical. Therefore, they should be put into one graph showing all three groups (“Glass or control”, “Plastic”, and “0 BPA”), and use one-way ANOVA for statistical analysis.
  6. Table 1 is unnecessary.
  7. Figure 3 and Figure 4, gene expression of the markers of macrophages, neutrophils, and T-cells did not mean too much without showing the number of infiltrating cells. Immunostaining of these markers in heart sections and quantification should be provided.
  8. For qRT-PCR, the primers for all genes examined should be provided. How was relative gene expression (RGE) calculated? A 2-(∆∆Ct) value should be used to present qRT-PCR data. It is also best to log transform the 2-(∆∆Ct) values before undertaking statistical analysis.
  9. Figure 5a represented the same data (cKit) as in Table 2. One of them should be removed.
  10. Figure 6, representative images should be provided for each area of each group (pericardial, myocardial, and vascular mast cells in Glass, Plastic, and 50 BPA). Also in Figure 7, representative images (NOT and Degran) of Glass, Plastic, and 50 BPA should be provided.
  11. Figure 8 is unnecessary. It showed the same graphs as in Figure 6 c&h and Figure 7 c&h.
  12. Figure 9, representative images should be provided for each area of each group (pericardial, myocardial, and vascular fibrosis in Glass, Plastic, and 50 BPA). Also, on Page 10, Line238, it should be Fig. 9g.

Author Response

The response letter is attached.

Reviewer 2 Report

The manuscript is about „Plastic caging alone increases acute myocarditis in male mice similar to high dose bisphenol A: but not through mast cell activation as in females“, brings new insight into the development of endocarditis in mice. Therefore it is suitable for publication in the current presentation!

Author Response

The response letter is attached.

Reviewer 3 Report

TO AUTHORS

The study examines the effects of plastic caging (that can be a source of potentially cardiotoxic endocrine disruptors) on myocardial inflammation and fibrosis in mice with viral myocarditis. The experiments are generally well-executed. The following concerns are raised: 

Major:

1.There is a disconnect between the inflammation and fibrosis data that the authors need to better discuss and attempt to explain. Plastic caging seems to increase myocardial inflammation only in males; however, fibrosis is increased only in females.

2.How do the data help in understanding the clinical observations on the sex-specific responses in myocarditis?

3.The title is convoluted and difficult to follow. Please simplify the title to indicate the main message of the study. For example: “Sex-specific effects of plastic caging in murine viral myocarditis”

4.Considering that the associations between BPA and cardiovascular disease involve primarily atherosclerotic and hypertensive heart disease (including MI(, the authors need to better explain the choice of a viral myocarditis model. Is there evidence that BPA affects the severity of myocarditis in human populations?

5.Please present data as box/scatter plots to appreciate variability.

6.Conclusions on macrophage/neutrophil/mast cell infiltration is based on assessment of gene expression. Do the authors have immunohistochemical data examining the effects of caging on leukocyte infiltration in the heart? Considering the availability of antibodies and histological material, this should be feasible.

7.Figure 6: Please provide representative images of mast cells. Also please define the units in the y axis (cells/mm2? Cells/hpf?)

8.Figure 9 legend: “Fibrosis was assessed as the % inflammation…” Please correct the error.

Author Response

The response letter is attached.

Round 2

Reviewer 1 Report

It is unacceptable to use the same data in two papers even though they are just a small portion compared to the whole manuscript and not as significant as the other data. Most data in that Table (Now Table 1 in the revised version, qRT-PCR results of CD45, CD11b, F4/80, CD3, CD4, and CD8) are identical to the same group's previous publication (same value, SEM, and p value). 

Author Response

Response: Thank you for your comment. We have now removed Table 1 from the paper and made reference to the data published in the previous paper within the results section of the text including the reference.

Reviewer 3 Report

The authors have addressed my concerns. I have no further recommendations.

Round 3

Reviewer 1 Report

The authors have made changes that improved the manuscript.